# The Potential of Food Fortification as an Enabler of More Environmentally Sustainable, Nutritionally Adequate Diets

**DOI:** 10.3390/nu15112473

**Published:** 2023-05-25

**Authors:** Alessandra C. Grasso, Julia J. F. Besselink, Marcelo Tyszler, Maaike J. Bruins

**Affiliations:** 1Blonk Sustainability Tools, 2805 TD Gouda, The Netherlands; 2dsm-firmenich, Taste, Texture and Health, 2613 AX Delft, The Netherlands; julia.besselink@dsm.com

**Keywords:** fortification, sustainable diets, plant-based, plant protein, plant substitute, plant alternative, diet optimization, micronutrients

## Abstract

Policies encouraging shifts towards more plant-based diets can lead to shortfalls in micronutrients typically present in animal products (B-vitamins, vitamin D, calcium, iodine, iron, selenium, zinc, and long-chain omega-3 fatty acids). We modelled the effect of fortifying foods with these critical micronutrients, with the aim of achieving nutrition and sustainability goals, using food consumption data from Dutch adults (19–30 years). Three dietary scenarios were optimized for nutritional adequacy and 2030 greenhouse gas emissions (GHGE-2030) targets, respectively, with the fewest deviations from the baseline diet: (i) the current diet (mainly vitamin A- and D-fortified margarine, iodized bread, and some calcium- and vitamin D-fortified dairy alternatives and iron- and vitamin B12-fortified meat alternatives); (ii) all plant-based alternatives fortified with critical micronutrients; and (iii) fortified bread and oils. Optimizing the current diet for nutrition and GHGE-2030 targets reduced animal-to-plant protein ratios from ~65:35, to 33:67 (women) and 20:80 (men), but required major increases in legumes and plant-based alternatives. When fortifying all plant-based alternatives and, subsequently, bread and oil, smaller dietary changes were needed to achieve nutrition and GHGE-2030 targets. Fortifying food products with critical micronutrients, ideally with complementary education on plant-based foods, can facilitate the transition to healthier and more sustainable diets.

## 1. Introduction

Current food systems are unsustainable [1,2] and responsible for 21–37% of global greenhouse gas emissions (GHGE) [3], depleting natural resources [3,4,5], and destroying biodiversity [6]. While diet-related chronic diseases are on the rise, almost one third of the world’s population lacks access to sufficient and nutritious food [7,8]. Therefore, the current food system needs to be redesigned to provide sustainable, nutritious, and affordable diets to a growing population [9]. To meet the goal of limiting global warming to 1.5 degrees, through GHGE reductions established by the UN’s Intergovernmental Panel on Climate Change (IPCC) [10], change is needed in food systems and diets. Increased availability of alternative (plant-based) proteins is one of the several actions proposed to help advance the European Union (EU) “Farm to Fork strategy” [11]. As a result, a shift towards more plant-based diets is warranted [12]. Consumer research shows that health and sustainability concerns are already driving consumers towards plant-based foods [13].

In comparison to foods from animal sources, plant-based foods such as legumes, nuts, and plant-based alternatives are typically low in saturated fats and rich in unsaturated fatty acids, dietary fiber, and phytonutrients such as polyphenols [14]. A (partial) replacement of animal products with plant-based foods may provide beneficial health outcomes, such as a reduced risk of type II diabetes, cancer, coronary heart disease, and premature mortality [12,15,16]. Plant-based foods also generally provide more thiamin, vitamin C, folate, and potassium than animal products [15]. On the other hand, animal foods are essential sources of macro- and micronutrients that are difficult to obtain from plant-based foods [14]. Plant-based foods and diets often have a lower quality [15] and content [16] of protein; choline; vitamins B2, B3, B5, B6, and D; iodine; selenium; and bioavailable calcium, iron, and zinc; and lack vitamin B12 and long-chain omega-3 polyunsaturated fatty acids, eicosapentaenoic, and docosahexaeonoic acids (EPA and DHA) [17,18,19,20,21,22,23]. Therefore, large shifts from animal- to plant-based diets will lead to nutritional advantages but also shortcomings [20,24,25]. Studies on vegetarians and vegans, for instance, show lower levels of vitamin B12, vitamin D, iron, zinc, iodine, calcium, selenium, and long-chain omega-3 fatty acids [23,26,27,28,29,30,31,32].

Fortification of foods with nutrients that are scarce or absent in plant-based diets can support the achievement of nutrient recommendations while switching to more plant-based dietary patterns [33,34,35,36]. Several modelling studies have predicted nutritional benefits but also shortcomings when shifting towards more plant-based diets [19,22,37]. Only a limited number of studies have included scenarios that might bridge these shortcomings in a sustainable way, to guide policy making [36,38]. However, in these studies, not all population nutrient recommendations were considered, and the number of fortified food options and fortified nutrients was limited.

In this study, different scenarios are modelled to explore how the fortification of plant-based alternatives and commonly eaten foods with critical nutrients could enable a transition towards more plant-based diets in an average diet of young Dutch adults. This research fills a knowledge gap on the role of fortified food in future-proof diets and the associated environmental impacts. It could also play a role in defining dietary strategies that fulfill both nutrient recommendations and the need to reduce GHGE to limit the global average temperature rise to 1.5 °C.

## 2. Materials and Methods

### 2.1. Diet Optimization Approach

Quadratic optimization modelling was conducted to model nutritionally adequate diets and nutritionally adequate diets that meet the 2030 GHGE target for different fortification scenarios. This modelling technique searches for a unique combination of food products to optimize a quadratic objective function (e.g., staying as close to the baseline diet as possible), while meeting a number of constraints or requirements (e.g., nutrient reference value, GHGE) [39]. The baseline diets, constraints, and steps are described in the following sections. Optimizations were performed using the diet optimization software, Optimeal 3.0^®^ (Blonk Consultants, Gouda, The Netherlands) [40]. Details of the optimization algorithm can be found in the Appendix A of Broekema et al. [41].

### 2.2. Baseline Diets

This study takes the average diets of Dutch men and women aged 19–30 years as baseline diets. Food consumption data were obtained from the Dutch National Food Consumption Survey (2007–2010) [42], which were processed in a previous study [43] and described in detail elsewhere [41]. In short, these data were collected from 3819 participants (*n* = 703 aged 19–30 years) by means of 2 24 h recalls on two non-consecutive days. Consumption of 1599 foods was reported and linked to the Dutch Food Composition Table 2016 [44]. The number of foods was reduced to 207 products based on their contribution to total consumed weight of the diet, availability of environmental data, and potential in more sustainable diets (e.g., plant-based meat and dairy alternatives). While these foods cover 77% of consumed weight and 56% of the energy intake across survey participants, the amount of the foods (in grams) was proportionally adjusted so that is represented close to 100% of the original energy intake. An average diet was derived by first averaging food items reported during the 2 recalls (in g/d) for each participant and then averaging food items across participants for men and women aged 19–30 years, respectively. For this analysis, an additional five meat and dairy alternative products from the Dutch Food Composition Table 2021 [45], and two fish alternative products (nutrient composition provided by dsm-firmenich) were added to the diet. Appendix A shows the detailed composition of the reference diets among Dutch women and men, of 19–30 years, used in this study.

### 2.3. Food Fortification Scenarios

This study explores three different fortification scenarios for the average diets of Dutch men and women aged 19–30: current diet (CUR); CUR with fortified meat, dairy, and fish alternatives (FORT-A); FORT-A with additionally fortified bread and oils (FORT-B).

CUR diet scenario: the CUR diet already includes some fortified food products, including breads with iodine, margarines with vitamins A and D, dairy alternatives with calcium, vitamins B2, B12, and D, and meat alternatives with iron and vitamin B12 (one with calcium, iron, and vitamin B6) (Appendix A).

FORT-A scenario: Plant-based dairy, meat, and fish alternatives were fortified with the long-chain omega-3 fatty acids (EPA and DHA), vitamins and minerals. These nutrients were selected because they were expected to become a concern when transitioning to a more plant-based diet. Meat, fish, and dairy alternatives were (per 100 g) fortified with vitamins and minerals at 30% of the Codex nutrient reference values (NRV) for adults in the EU (Appendix A) as a reasonable amount to bridge the difference and for manufacturers to allow for a “high in” claim on the product [44]. Meat and fish alternatives were fortified with selenium and vitamins B2, B3, B6, and B12. Vitamin B5 was not taken into consideration as no data were available from the NEVO database. In addition, meat alternatives were fortified with iron and zinc and fish alternatives with vitamin D3, iodine, and DHA and EPA. Dairy alternatives were fortified with vitamin B2, B12, D3, calcium, selenium, iodine, and zinc. Fish alternatives were (per 100 g) fortified with 80 mg of DHA and EPA (32% of the EU AI) to bridge the gap with fish and allow for a “high in omega-3” claim.

FORT-B scenario: Additionally, all bread and oils were fortified with vitamins and minerals at 15% of the Codex NRV, namely DHA and EPA, calcium, iron, zinc, selenium, iodine, and vitamins B2, B3, B6, B12, and D3. Since these foods were not designed to replace animal products and are consumed by the whole population, they were fortified at lower levels (15% of the EU NRV) than the plant-based alternatives. Bread and oils were (per 100 g) fortified, with 40 mg of DHA and EPA (16% of the EU AI) to allow for “a source of omega-3” claim. Appendix A summarizes the nutrients, fortified products, and fortified amounts. Overages for vitamins were considered to compensate for potential losses during processing and shelf life.

### 2.4. Nutritional Constraints

Population reference intake (PRI) values set by the Health Council of the Netherlands or taken from international recommendations were used to derive lower limits of nutritional reference intakes [46,47,48,49,50]. If values for PRI were not available, values for adequate intake were used. The acceptable macronutrient distribution ranges were used to derive lower and upper limits for protein, carbohydrates, and fatty acids [49,51]. Upper limits of nutrients were based on values for the tolerable upper intake levels [49,50]. Appendix A lists these nutritional constraints.

### 2.5. Environmental Impact Data and Food System GHGE Target

The environmental impact was measured using life cycle assessments (LCA), a methodology that accounts for the environmental impact across the life cycle of a food product, including all activities taking place at the farm all the way through processing, retail, consumption, and food loss and waste. The geographical scope of the LCAs is the Netherlands. The environmental impact of each food item was assessed using three environmental impact indicators, namely GHGE (kg CO_2_ eq), land occupation (m^2^*y), and blue water use (m^3^), and were calculated by applying the ReCiPe Midpoint 2016 method [52]. The environmental impacts of the food products were reported in a previous study [41]. The environmental impacts of the additional meat and dairy alternatives were taken from the Dutch National Institute for Public Health and the Environment’s LCA database [53]. The environmental impacts of the added fish alternatives were calculated based on extrapolation from a recipe provided by dsm-firmenich. The GHGE of the fortified foods were adapted to account for the addition of maltodextrin powder blended with vitamins and minerals and algal-derived DHA and EPA (land use and water consumption were not).

We considered a 2030 food system GHGE target of 2.04 kg CO_2_-eq per person per day, which Broekema et al. [41] derived from the IPCC 1.5 °C assessment study [10]. This reduction target, focusing on food consumption in the Netherlands, was set to limit global average temperature rise to 1.5 °C [41]. The details about the calculation can be found in the Appendix A of Broekema et al. [41].

### 2.6. Food Product Constraints

To prevent extremely large changes for single food products, a maximum constraint was placed on the quantity of each food product [43]. Because a proxy to an acceptable diet is one with minimum deviation from the baseline diet, additional constraints on the food groups were implemented equal to 10–500% of current intake in a secondary analysis. A narrower range led to no solution for at least one of the optimized scenarios.

### 2.7. Optimization Strategy

Firstly, the current and fortified dietary scenarios were optimized for nutritional adequacy, which was defined by the fulfillment of a set of daily energy and nutrient requirements described earlier. Secondly, the scenarios were optimized for the 2030 GHGE target of 2.04 kg CO_2_-eq/d. The deviation from current and fortified diets to the optimized diets, measured as the Euclidean distance, was examined to determine how different the optimized diets are to the baseline diet. Diets similar to the baseline diet in terms of diet composition, i.e., with a smaller Euclidean distance, were considered more culturally acceptable, whereas diets with larger Euclidean distance were considered to have greater risk of lower acceptability [54].

## 3. Results

Figure 1 and Figure 2 show how the animal-to-plant (A:P) protein ratio and the composition of the diet of Dutch women and men aged 19–30 years changed for the three fortification scenarios, first after optimizing for nutritional adequacy and then for the 2030 GHGE targets. The results in each step are described in detail below. Appendix A shows the same scenarios but implementing acceptability constraints on food groups of 10–500% of current intake.

### 3.1. Baseline Diets (before Optimization)

The baseline diets had an A:P protein ratio of about 65:35 (Figure 1). The current (CUR) baseline diet had a GHGE of 4.24 kg CO_2_-eq/d (women) and 5.73 kg CO_2_-eq/d (men) (Table 1a,b). The GHGE of the baseline diet with fortified plant-based alternatives (FORT-A) and with additional fortification of oils and breads (FORT-B) was 0.7% higher, compared to the CUR diet. The baseline of the current diet scenario fell short of several nutrient recommendations for women and men (see Table 1a,b and Appendix A for a complete overview of nutrient intake). Intakes of fiber, vitamins A, B1, B2, B6, D, folate, potassium, magnesium, iron, selenium, α-linolenic acid, and DHA and EPA were lower than recommended for women. Intakes of fiber, vitamins A, B1, and D, folate, selenium, α-linolenic acid, and DHA and EPA were lower than recommended for men. Moreover, the baseline diet was high in saturated fats and sodium (men only). The baseline diet of the FORT-A scenario had very minor increases in the nutrients that were fortified and remained short of the same nutrients. In the baseline diet of the FORT-B scenario, the women’s diet was no longer short of vitamins B2 and B6, and DHA and EPA, and the men’s diet was no longer short of selenium and DHA and EPA (Appendix A). Replacing oils and breads with their fortified counterpart led to substantial increases in intakes of vitamin D and iron for both men and women, but these still fell below recommendations.

**Figure 1 nutrients-15-02473-f001:**
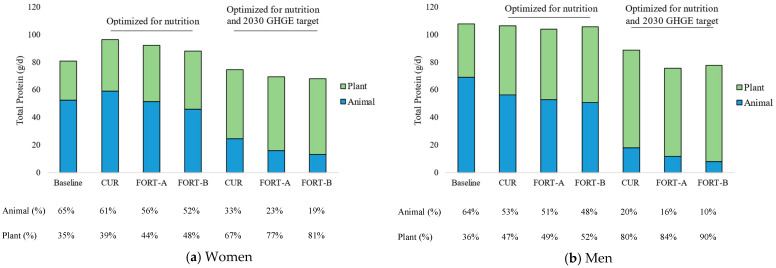
Contribution of animal and plant protein to total protein intake (g/d) among Dutch women (**a**) and men (**b**) aged 19–30 years for three fortification scenarios: current diet (CUR); CUR with fortified meat, dairy, and fish alternatives (FORT-A); FORT-A with fortified bread and oils (FORT-B), optimized for nutritional adequacy and 2030 greenhouse gas emission (GHGE) target.

### 3.2. Nutritionally Optimized Diets

When the baseline diets were optimized for nutrition, total protein intake increased for women between 9–19%, with the highest increase in the CUR scenario and the lowest in the FORT-B scenario, and decreased for men between 1–3% (Figure 1).

In the scenario where the CUR diet was optimized for nutrition, GHGE increased by 12% for women and decreased by 11% for men. While the A:P protein intake ratio decreased for both women and men, the relative contribution of animal protein was higher for women compared to men (61% vs. 53%).

To meet the nutritional recommendations, various changes had to be made to the composition of the diet (Figure 2 and Appendix A). In general, there was a substantial increase in the amount of vegetables, legumes, meat, dairy, and fish alternatives, fish and fish products, eggs and egg products, and fats and oils, and a reduction in meat and meat products (23% for women and 56% for men) and discretionary food products. Due to women’s higher requirement for iron, meat decreased less, and fish increased more, compared to the diet of men. Fish increased more than 6-fold for women and 4-fold for men, to meet the recommendations for selenium and vitamins B2 and D, reaching the upper limit of DHA-EPA of 1000 mg/d.

**Figure 2 nutrients-15-02473-f002:**
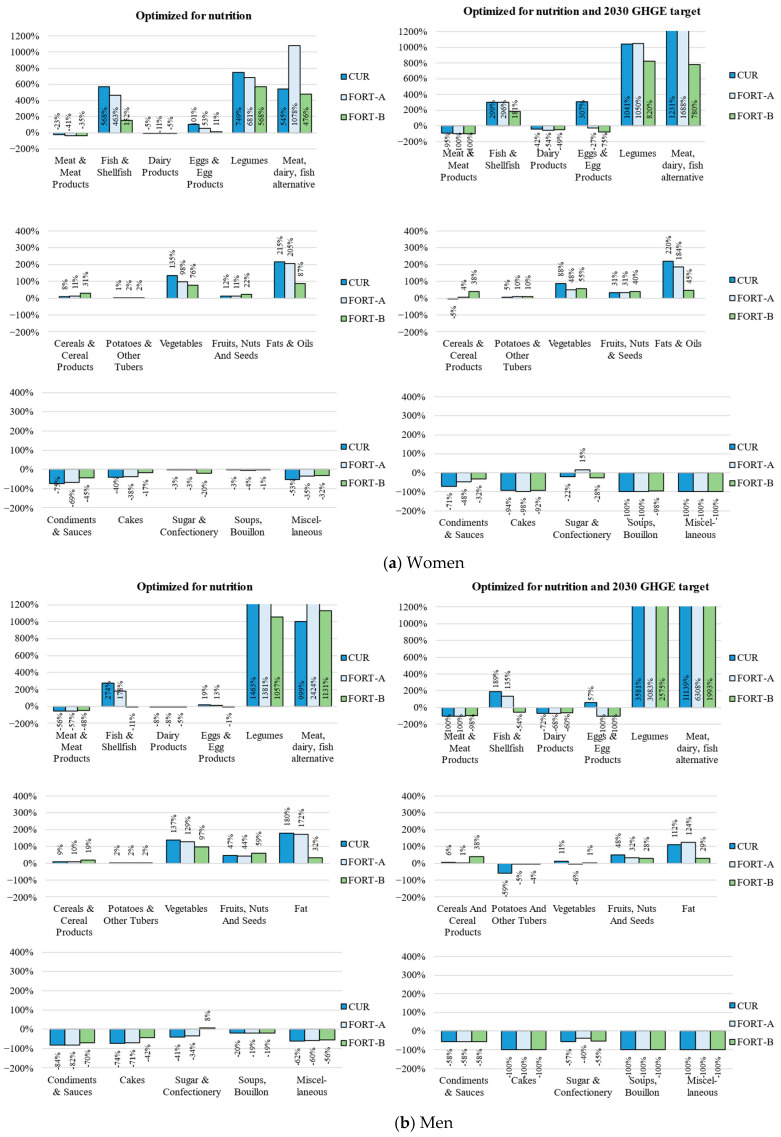
Percent change in consumed quantity of food groups relative to baseline diet among Dutch women (**a**) and men (**b**) aged 19–30 years for three fortification scenarios: current diet (CUR); CUR with fortified meat, dairy, and fish alternatives (FORT-A); FORT-A with fortified bread and oils (FORT-B), optimized for nutritional adequacy and 2030 greenhouse gas emission (GHGE) target.

The relative contribution of animal-to-total protein was lower in the diets with more fortified food products when optimized for nutrition (Figure 1). The A:P protein decreased from 61:39 to 56:44 (women) and from 53:47 to 51:49 (men) in the FORT-A scenario (fortified meat and fish alternatives), and to 52:48 (women) and 48:52 (men) in the FORT-B scenario (additionally fortified bread and oil). The GHGE of the diets was also lower in the scenarios with more fortified food products.

To achieve nutrition targets, adding fortified plant-based alternatives to the diet (FORT-A) reduced the required deviation from the baseline diet compared to the CUR scenario (Euclidean distance = 66.5 vs. 75.2 for women (Table 1a) and 76.7 vs. 80.5 for men (Table 1b)). Furthermore, the magnitude of the changes required for the diet decreased, except for plant-based alternatives in the FORT-A scenario (Figure 2). That is because plant-based alternatives became one of the most important contributors of calcium, zinc, selenium, iron, and vitamins B2, B6, and D. To achieve the nutritional recommendations (FORT-A), the amount of dairy alternatives increased from 6 to 32 g/d (women) and from 2 to 9 g/d (men), meat alternatives from 1 to 37 g/d (women) and 0.3 to 9 g/d (men), and fish alternatives from 0 to 37 g/d (women) and 0 to 5 g/d (men). When optimizing the diet with added fortified bread and oils (FORT-B), the deviation from the baseline diet was even smaller (Euclidean distance = 42.1 for women (Table 1a) and 5.7 for men (Table 1b)), and smaller changes in the food groups were required, except for minor increases in change in cereals and cereal products and fruits, nuts, and seeds. Cereals became the most important contributor of selenium and vitamins B1, B6, and D.

When optimizing the CUR baseline diet of women (Table 1a) and men (Table 1b) for nutrition recommendations, the amount of polyunsaturated fats, fiber, vitamins, minerals (except calcium), and EPA and DHA had to be increased. Women had to increase protein intake while men had to reduce sodium intake. When including fortified plant-based alternatives (FORT-A) as well as fortified bread and oil (FORT-B), nutrient intakes had to be further increased, but to a lesser extent than with the CUR diet.

### 3.3. Diets Optimized for Nutrition and Meeting 2030 GHGE Targets

To meet the nutritional recommendations and reduce GHGE to 2.04 kg CO_2_-eq/day with the current diet, meat was reduced to reach a A:P ratio of 33:67 (women) and 20:80 (men) (Figure 1). The amount of fish and fish products increased 4-fold for women and almost 3-fold for men in the CUR scenario. Eggs also increased 4-fold for women, and only by about 50% for men. The amount of legumes and plant-based alternatives increased by >1000%. In the CUR scenario, the increase in plant-based alternatives was due to increases (1450% for women and 12,930% for men) in plant-based dairy alternatives which were already fortified with fiber, calcium, and vitamins B2, B12, and D. The diet of men had a much greater increase in dairy alternatives, to compensate for the greater reduction in calcium and vitamin B2 from dairy products and vitamin B12 from meat products. The other plant-based alternatives were not in the optimized diet. When increases in food groups were capped at 500% of current consumption, there were slightly larger increases in fish and fish products, eggs, and vegetables in the CUR scenario (Appendix A).

When introducing fortified plant-based alternatives (FORT-A), the deviation from the baseline diet was smaller compared to the CUR scenario (Euclidean distance = 141.6 vs. 194.9 for women and 266.8 vs. 383.3 for men (Table 1a,b)). Moreover, the magnitude of changes in food groups to achieve nutrition and GHGE targets became smaller, except for a slight increase in legumes and plant-based alternatives for women, but the change in the A:P ratio became larger, with a shift to an A:P ratio of 23:77 (women) and 16:84 (men). The increase in the amount of plant-based alternatives was largely due to an increase in dairy alternatives (1280% for women and 6800% for men) and an introduction of fish alternatives to the diet (26 g/d for women and 9 g/d for men). Meat alternatives also increased in the diet of the women by 1030%, but were removed from the diet of the men. When also including fortified bread and oil (FORT-B), the deviation from the baseline diet was even smaller (Euclidean distance = 114.3 for women and 223.3 for men (Table 1a,b)) Additionally, the changes in food groups (Figure 2) led to a greater change in the A:P ratio—19:81 and 10:90 for women and men, respectively (Figure 1).

Progressively optimizing for nutrition and GHGE-2030 goals reduced protein intake for women by 7–16%, from 81 g with the baseline diet to 68–75 g with the nutrition and GHGE-2030 optimized diet, in all the fortification scenarios (Table 1a). For men, protein intakes fell by 17–29%, from 107 g to 76–78 g in the different scenarios (Table 1b). The intakes of most of the micronutrients increased when optimizing for nutrition and GHGE-2030, as compared to the baseline diet. Energy and micronutrient intakes were slightly lower than in the nutrition-optimized diet.

## 4. Discussion

The current study shows that fortification of plant-based alternatives to meat, dairy, and fish, and of commonly eaten foods (e.g., bread, oil) with critical nutrients, could concomitantly improve nutritional adequacy and lower GHGE, compared to the current diet and with fewer changes to it.

Currently, Dutch people consume large amounts of dairy and meat products, while the consumption of legumes, fruit, vegetables, and nuts is insufficient [42]. Despite some of the foods in the Dutch diet being fortified (mainly vitamin A- and D-fortified margarine, iodized bread, and some dairy and meat alternatives) some vitamin and mineral inadequacies still exist [55]. To achieve the GHGE-2030 targets with the current diet, in addition to nutrition targets, larger consumption was needed for legumes and plant-based alternatives, and lower total protein consumption and A:P protein intake ratios.

Several modelling studies have predicted nutritional shortcomings when shifting towards more plant-based diets [19,22,37]. One simulation study of the Dutch diet, for instance, showed that a 100% replacement of unfortified meat and dairy with plant-based alternatives would reduce the environmental impact by >40%, but also compromise intakes of calcium, iron, zinc, thiamin, vitamin B12, and vitamin A [56]. On the other hand, fortifying 30% of meat alternatives with iron and vitamin B12, and dairy alternatives with calcium, riboflavin, and vitamin D was beneficial for saturated fatty acids, sodium, fiber, and vitamin D intakes, and led to a 14% reduction in environmental impact [56]. A simulation study using a dataset of four European diets showed that fully optimizing these diets for both nutrition and sustainability, within acceptable degrees of dietary change, could not be achieved, i.e., intakes of fiber, potassium, magnesium, and vitamin D and E were still below recommendations [57]. The authors optimized the same diets for dietary guidelines and, subsequently, for (1) preferences, (2) nutrient quality (nutrient-rich diet score 15.3, i.e., 3 nutrients to limit and 15 to encourage), or (3) GHGE targets [36]. For all optimizations, meat had to be reduced, nutrition scores had to be improved, and GHGE had to be reduced. Compared to the diet scenario without fortification, a further 3% reduction of GHGE could be achieved if meat alternatives were fortified with iron, vitamin B1, B2, B3, and B12 for all optimizations. A recent modelling study of French diets found that when optimizing diets for nutrition, plant-based substitutes fortified with iron and zinc allowed for greater red meat reductions, with minimal deviations from the current diet for other food groups [58]. Another modelling study focusing on vitamin D as the first limiting nutrient, showed that achieving sufficient vitamin D intake within environmental limits is only possible with a shift away from meat and dairy, more fish, and the inclusion of more vitamin D-fortified foods in an average Dutch diet [38].

Fish-derived long-chain omega-3 (EPA and DHA) intake is important to support heart, brain, and eye health. Nevertheless, Europeans are not meeting the recommended intake of 250 mg/d [21]. So far, no simulation studies have introduced solutions to counteract further long-chain omega-3 shortfalls when shifting to more plant-based diets. While various simulation studies showed that diets can be optimized for both nutrition and GHGE by introducing nutrient-enriched plant-based substitutes, this is the first study showing that introducing complementary algal-derived DHA- and EPA-fortified foods can bridge potential gaps caused by a reduction in fish omega-3 intake.

Another unintended consequence of optimizing for nutrition and GHGE-2030 targets was the reduced average protein intake to 68 g/d for women and 78 g/d for men. Considering the lower protein quality of plant protein compared to animal protein sources, protein intake may become critical for those relying on one or few protein sources [15]. A modelling study of Dutch diets showed that cutting down on animal protein to reduce diet-related GHGE is feasible for reductions of up to 12–16% without compromising protein adequacy and quality. Another modelling study of French diets showed that, if more than 50% and 70% of animal sources are replaced by plant sources, protein quantity and quality may become limited, respectively [59].

This study has several strengths and limitations. A strength of this study is that it optimizes both nutrition and sustainability, and addresses the nutrients of concern that may fall short when shifting towards a more plant-based diet, through a realistic solution. Compared with other modelling studies, we considered three environmental impact indicators and used environmental impact data that were consistently calculated over the entire life cycle of the product. Furthermore, the analysis serially accounted for nutrition and climate. While the model was set to find solutions as close as possible to the baseline diet, they may prove difficult to reach in practice, as food groups were removed from the diet, such as meat and meat products, and consumption of two other groups—legumes and plant-based alternatives—increased by >1000%. In the most extreme case, when the CUR diet for men was optimized for nutrition and 2030 GHGE targets, plant-based alternatives increased by approximately 10,000%, i.e., from 2 to 233 g/d for men, which is equivalent to a glass of soy milk a day. In the same scenario, legumes increased by about 3600%, i.e., from 3.3 to 120 g/d, which is equivalent to a half can of kidney beans per day [60]. Another study, in which diets of Swedish adolescents were optimized for nutritional adequacy and sustainability, also found that significant changes in food group consumption were needed, with particularly large increases for fortified plant-based substitutes, with the vegan scenario having greatest reduction in GHGE [61].

Because the current study was primarily exploratory, additional acceptability constraints, such as a minimum quantity of food groups, were not established in the primary results. Although the food consumption survey used in this study was from 2007–2010, and was applied to a narrow age group (19–30 years), it is likely sufficient to illustrate the concept of introducing fortification, in relation to nutrition and GHGE targets. The foods were fortified at 30% of the NRV, to largely bridge the gap with meat, fish, or dairy alternatives, but, in the future, a more precise approach might be considered. In our model we did not consider the bioavailability of minerals that are likely to be lower in plant than in animal sources. The bioavailability of minerals is difficult to quantify, as it depends on many factors, including inhibitors and enhancers in the food, and background diet (e.g., the presence of anti-nutrients such as phytic acid or tannins), the mineral form used to fortify, and the mineral status of the individual. In addition, the current model optimizes for the average intake of the population, at the recommended intake, but, due to distribution, not all individuals may succeed in achieving this goal.

Plant-based diets have been demonstrated to have nutritional and environmental advantages, although data also suggest that shortfalls might appear when progressively restricting animal foods [24,25,62]. Whereas nearly half of the national dietary guidelines mention plant-based alternatives, they are generally not treated as an alternative to meat or dairy in terms of nutrition [62]. Codex guidelines recommend that, where a substitute food is intended to replace a food, nutrients need to be restored to levels that are nutritionally equivalent [63]. However, in Europe, only 13% of the meat and 26% of the dairy alternative products introduced between 2020–2022 were fortified with vitamin B12 [64]. Consuming foods made from plants is becoming more popular; however, the high variability in the nutrient quality of these plant-based alternatives points to the need for further education and nutrition standards. Simulation of diets that include solutions that enable both nutrition and environmental improvement can guide new nutrition guidelines and help consumers make the right dietary choices. The current simulation study shows the potential of food fortification as an enabler of more environmentally sustainable, nutritionally adequate plant-based diets. Because of the negligible amounts added to foods, micronutrients have a negligible impact on sustainability, and a large impact on nutrition. Future research could investigate not only the fortification of foods, but several complementary interventions that could enable shifts towards nutritious and sustainable diets, such as solutions that contribute to sustainable animal product production, alternative protein sources, more bioavailable minerals, iron, zinc, and β-carotene-biofortified crops, and eggs and milk enriched with micronutrients and DHA and EPA through feed fortification. Fortifying (plant-based) food products with essential micronutrients, ideally in conjunction with other solutions and education on plant-based nutrition, can facilitate the transition to healthier and more sustainable diets.

## Figures and Tables

**Table 1 nutrients-15-02473-t001:** Daily nutrient intake, environmental impact and Euclidean distance of Dutch (**a**) women and (**b**) men aged 19–30 years for three fortification scenarios: current diet (CUR); CUR with fortified meat, dairy, and fish alternatives (FORT-A); FORT-A with fortified bread and oils (FORT-B).

	(a) WOMEN									
		Baseline Diet			Optimized for Nutritional Adequacy			Optimized for Nutritional Adequacy + 2030 GHGE Targets		
	DRV	CUR	FORT-A	FORT-B	CUR	FORT-A	FORT-B	CUR	FORT-A	FORT-B
Energy (kcal)	2220	2024	2024	2024	2216	2216	2214	2047	2071	1981
Protein total (g)	52	81	81	81	96	92	88	75	69	68
Fat total (g)	93	80	80	80	90	90	83	93	89	70
SAFA (g)	23	28	28	28	23	23	23	23	20	15
PUFA (g)	28	16	16	16	28	28	22	26	28	21
Linoleic acid (g)	4.7	13.3	13.3	13.3	22.4	22.3	18.0	22.4	23.5	17.6
α-Linolenic acid (g)	2.3	1.6	1.6	1.6	3.9	3.8	2.8	2.9	3.2	2.3
Trans fatty acids (g)	2.3	1.0	1.0	1.0	1.1	1.1	1.0	0.6	0.5	0.3
Carbohydrates (g)	210	231	231	231	236	240	260	210	228	249
Fiber (g)	29	18	18	18	29	29	29	30	32	33
Water (g)	2300	2953	2953	2953	3205	3143	3087	2754	2792	2782
Alcohol (g)	10.0	1.7	1.7	1.7	1.7	1.6	1.7	0.9	1.3	1.4
Retinol act. eq. (μg)	680	574	574	574	1571	1075	841	1395	844	680
Vitamin B1 (mg)	0.88	0.72	0.72	0.72	0.91	0.88	0.88	0.88	0.88	0.88
Folate eq. (μg)	300	218	218	218	332	316	300	325	300	300
Vitamin B2 (mg)	1.60	1.27	1.30	1.63	1.6	1.8	2.0	1.6	1.6	1.8
Niacin (mg)	14	16.0	16.1	16.4	22	20	19	19	19	19
Vitamin B6 (mg)	1.5	1.3	1.4	1.7	1.8	1.9	2.1	1.5	1.5	2.0
Vitamin B12 (μg)	2.8	4.08	4.10	4.71	8.6	7.3	5.4	6.3	5.5	4.7
Vitamin D (μg)	10.0	2.5	2.6	5.8	10.0	10.0	10.0	10.0	10.0	10.0
Iodine (μg)	150	163	166	202	199	210	260	182	211	276
Calcium (mg)	950	1021	1036	1226	1014	1061	1326	950	950	1069
Iron (mg)	16.0	8.9	8.9	12.3	16.0	16.0	17.1	16.0	16.0	18.9
Selenium (μg)	70	42	43	55	73	80	74	70	70	70
Zinc (mg)	7.0	10.2	10.4	12.9	11.4	13.1	15.2	9.5	11.0	13.1
DHA + EPA (mg)	200	141	141	205	1000	1000	593	1000	1000	974
Vitamin C (mg)	75	75	75	75	114	101	100	109	85	85
Vitamin E (mg)	11	13	13	13	19	19	16	19	18	13
Vitamin K (μg)	90	113	113	113	283	225	191	393	138	126
Phosphorus (mg)	550	1484	1484	1484	1726	1653	1627	1557	1365	1339
Sodium (mg)	2400	2387	2387	2387	2400	2400	2400	1919	1910	1993
Potassium (mg)	3500	2971	2971	2971	3954	3769	3531	3500	3500	3500
Magnesium (mg)	300	299	299	299	396	396	381	479	476	449
Copper (mg)	0.9	1.0	1.0	1.0	1.5	1.5	1.4	1.8	1.9	1.8
Cholesterol (mg)		190	190	190	223	182	159	246	80	53
Global warming (kg CO_2_eq) ^1^		4.24	4.27	4.27	4.7	4.4	4.1	2.0	2.0	2.0
Land use (m^2^*y) ^2^		2.89	2.89	2.78	3.0	2.9	2.8	2.2	2.0	1.8
Water consumption (m^3^)		0.17	0.18	0.18	0.18	0.18	0.18	0.14	0.16	0.17
Euclidean distance ^3^		-	-	-	75.2	66.5	42.1	194.9	141.6	114.3
	**(b) MEN**									
		**Baseline Diet**				**Optimized for Nutritional Adequacy**		**Optimized for Nutritional Adequacy + 2030 GHGE Target**		
	**DRV**	**CUR**	**FORT-A**	**FORT-B**	**CUR**	**FORT-A**	**FORT-B**	**CUR**	**FORT-A**	**FORT-B**
Energy (kcal)	2850	2832	2832	2832	2847	2847	2842	2550	2550	2550
Protein total (g)	61	107	107	107	106	104	105	89	76	78
Fat total (g)	120	109	109	109	113	112	97	103	98	80
SAFA (g)	23	37	37	37	30	30	29	21	20	16
PUFA (g)	36	22	22	22	35	34	25	32	32	28
Linoleic acid (g)	6.0	18.9	18.9	18.9	27.9	27.6	21.0	27.3	27.1	25.1
α-Linolenic acid (g)	3.0	2.5	2.5	2.5	4.5	4.4	3.0	3.8	3.5	3.0
Trans fatty acids (g)	3.0	1.2	1.2	1.2	1.1	1.1	0.9	0.6	0.6	0.4
Carbohydrates (g)	473	309	309	309	299	302	334	270	291	328
Fiber (g)	36	23	23	23	36	36	36	40	39	42
Water (g)	2500	3134	3134	3134	3301	3300	3235	2500	2500	2500
Alcohol (g)	20.0	18.7	18.7	18.7	18.7	18.7	18.7	15.1	16.9	17.2
Retinol act. eq. (μg)	800	723	723	723	1308	1277	890	800	800	800
Vitamin B1 (mg)	1	0.98	0.98	0.98	1.13	1.13	1.13	1.13	1.13	1.13
Folate eq. (μg)	300	275	275	275	384	379	368	378	321	342
Vitamin B2 (mg)	1.60	1.67	1.68	2.15	1.7	1.9	2.3	1.6	1.7	2.0
Niacin (mg)	18	26.4	26.4	27.0	28	27	27	25	24	24
Vitamin B6 (mg)	1.5	1.8	1.8	2.3	2.0	2.1	2.6	1.5	1.5	2.1
Vitamin B12 (μg)	2.8	5.3	5.2	6.1	6.1	5.7	5.2	5.9	4.6	3.1
Vitamin D (μg)	10.0	3.5	3.5	8.0	10.0	10.0	10.0	10.0	10.0	10.0
Iodine (μg)	150	210	210	260	224	236	287	228	229	328
Calcium (mg)	950	1180	1185	1450	1086	1170	1474	950	1010	1179
Iron (mg)	11.0	11.16	11.18	15.95	13.7	14.0	20.0	14.2	12.2	19.8
Selenium (μg)	70	57.1	57.5	73.9	70	70	75	70	70	70
Zinc (mg)	9.0	13.3	13.4	16.8	12.4	13.3	17.7	10.3	12.6	15.7
DHA + EPA (mg)	200	119	119	207	999	820	394	1000	924	487
Vitamin C (mg)	75	84	84	84	127	124	113	90	75	75
Vitamin E (mg)	13	16	16	16	22	22	17	18	19	16
Vitamin K (μg)	120	133	133	133	279	270	235	165	120	133
Phosphorus (mg)	550	1969	1969	1969	2006	1973	2028	1762	1568	1607
Sodium (mg)	2400	3388	3388	3388	2400	2400	2400	2400	2020	2087
Potassium (mg)	3500	3904	3904	3904	4509	4465	4474	3710	3500	3500
Magnesium (mg)	350	394	394	394	475	476	503	600	523	518
Copper (mg)	0.9	1.2	1.2	1.2	1.6	1.7	1.8	2.1	1.8	1.8
Cholesterol (mg)		247	247	247	194	181	170	148	42	31
Global warming (kg CO_2_eq) ^1^		5.73	5.74	5.74	5.06	4.97	4.94	2.04	2.04	2.04
Land use (m^2^*y) ^2^		3.82	3.82	3.82	3.46	3.45	3.46	2.16	2.03	1.90
Water consumption (m^3^)		0.18	0.18	0.18	0.18	0.18	0.21	0.16	0.11	0.12
Euclidean distance ^3^		-	-	-	80.5	76.7	5.7	383.8	266.8	223.3

^1^ excluding land use change, ^2^ square meter years, ^3^ higher is more difficult/less acceptable.

## Data Availability

Data are contained within the article or Appendix A. The data presented in this study are available in Appendix A.

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
