# Peer review of "The Potential of Food Fortification as an Enabler of More Environmentally Sustainable, Nutritionally Adequate Diets"

_nutrients, 2023, doi:10.3390/nu15112473_

Round 1

Reviewer 1 Report

This was a well-written manuscript that focused on diet scenarios to not only help a population meet requirements, but also reduce gas emissions. It is an extremely timely topic. Some suggestions, though, are provided to further strengthen this manuscript.

There were a number of figures included as supplements and unsure the rationale for limiting the figures within the main text. For example, the Dutch recommendations may have been suited within the text as opposed to as a supplement to showcase what people are consuming compared to the recommendations in a numerical format as opposed to a figure format as displayed in the manuscript.

Abstract: May suggest highlighting the types of micronutrients to be found in fortified foods as may have different fortification policies depending on the country. For the final conclusive sentence, may consider placing implications of how this information can be used moving forward – education about consuming more plant-based diets, etc.

Introduction:

The first paragraph was lengthy, may considering adding a paragraph when explaining more about the benefits of consuming plant-based diets.

As the focus is on the Dutch diet, may consider providing some context about the dietary pattern generally consumed and the types of foods generally fortified and with the types of nutrients.

Methods:

For the age range, it is only from 19-30 years, as this is a tight age range, possibly including in the introduction or even the methods this rationalization.

The Diet optimization approach may need to be closer to what would be considered a statistical analysis approach as the way it is worded includes the software and such to determine these findings.

In the optimization strategy section, mentions smaller or larger distance, but not exact values. Are there ranges to determine if it is smaller or larger for statistical purposes? If so, please include that information.

In figure 2, unsure why there is percentages in certain food groups unless to demonstrate how much to increase/decrease consumption by, but if that is the case, then need to include it for the other foods listed.

Discussion:

The discussion is quite limited in discussing the results. There was rich insight to these modeling systems that could have been further mentioned as was indicated in the introduction. May consider then moving some information from the introduction to elaborate on the discussion.

Author Response

This was a well-written manuscript that focused on diet scenarios to not only help a population meet requirements, but also reduce gas emissions. It is an extremely timely topic. Some suggestions, though, are provided to further strengthen this manuscript.

We thank the reviewer for taking the time to review our manuscript and for his/her valuable suggestions.

There were a number of figures included as supplements and unsure the rationale for limiting the figures within the main text. For example, the Dutch recommendations may have been suited within the text as opposed to as a supplement to showcase what people are consuming compared to the recommendations in a numerical format as opposed to a figure format as displayed in the manuscript.

We have included an additional Table in the main text, showing how nutrient intakes would shift when optimized for nutritional adequacy and 2030 greenhouse gas emissions (GHGE-2030) targets, respectively.

Abstract: May suggest highlighting the types of micronutrients to be found in fortified foods as may have different fortification policies depending on the country. For the final conclusive sentence, may consider placing implications of how this information can be used moving forward – education about consuming more plant-based diets, etc.

We have specified the micronutrients in the abstract (line 10-11). In the final sentence of the abstract (line 22-24) and discussion (line 446-449), we summarized how this study can be useful and what else is needed to facilitate the transition to healthy and sustainable diets.

Introduction:

The first paragraph was lengthy, may considering adding a paragraph when explaining more about the benefits of consuming plant-based diets.

The authors have now significantly shortened the introduction and balanced information related to benefits and disadvantages of plant-based diets.

As the focus is on the Dutch diet, may consider providing some context about the dietary pattern generally consumed and the types of foods generally fortified and with the types of nutrients.

We agree with the reviewer that some background on the Dutch diet was missing. We have provided more context in the abstract (line 15-17), and discussion (line 242-245) about the consumed food groups, and about typically fortified foods in the Netherlands.

Methods:

For the age range, it is only from 19-30 years, as this is a tight age range, possibly including in the introduction or even the methods this rationalization.

The age range is currently only mentioned in the methods (line 112). We agree with the reviewer that this tight age range should be more emphasized; and have clarified in the introduction (line 94) and discussion (line 415).

The Diet optimization approach may need to be closer to what would be considered a statistical analysis approach as the way it is worded includes the software and such to determine these findings.

The diet optimization is not a statistical analysis, but a model-based mathematical optimization analysis. It optimizes a single average reference diet and as such does not contain statistical information on actual consumption, besides the starting/reference point, to which the optimization algorithm has as goal to stay as close as possible. One could add simulations with noise to the data underlying the optimization procedure to gather insights into uncertainty and range, but this was beyond scope of this current study. Details of the optimization algorithm has been published elsewhere [1] (supplementary materials) and a direct reference to it has been added to the manuscript.

In the optimization strategy section, mentions smaller or larger distance, but not exact values. Are there ranges to determine if it is smaller or larger for statistical purposes? If so, please include that information.

Since the optimization algorithms tries to find a diet as close as possible to the original diet, the difference between the optimized diet and the original diet can be measured as the Euclidean distance. The multidimensional Euclidean distance is the equivalent of the of the distance between 2 points (a line) but is applied here to measure the distance/difference between two baskets of products. For each optimized diet there is a resulting Euclidean distance. In the analysis we mentioned that diets with smaller distance, i.e. more similar to the baseline diet, are considered more acceptable than diets with larger distances, i.e. more different than the starting diet. The exact values are mentioned in Table 1 (last row). There is no statistical component to determine if a distance is in an acceptable range, since the level of the distance is influenced by the amount of products. The Euclidean distance serves as a ceteris paribus between diet comparison, where you can say that diets with a lower Euclidean distance is more likely to be acceptable than diets with a large Distance. In that sense, the diet FORT-B is the most acceptable since the distance is the smallest, but adding an environmental constraint increases the distance since more changes are required.

In figure 2, unsure why there is percentages in certain food groups unless to demonstrate how much to increase/decrease consumption by, but if that is the case, then need to include it for the other foods listed.

We had included values for the bars that exceeded the y-axis, as no Y-axis break was possible. We agree that it’s better to show data labels for all food groups.

Discussion:

The discussion is quite limited in discussing the results. There was rich insight to these modeling systems that could have been further mentioned as was indicated in the introduction. May consider then moving some information from the introduction to elaborate on the discussion.

We thank the reviewer for his/her suggestions and have now moved some information to the discussion and elaborated on some of the results of similar modelling systems throughout the discussion (see track changes).

Reviewer 2 Report

This work examines food fortification of a plant-based food supply to improve diet sustainability while maintaining nutritional quality of the diet.  The paper is well written and outlines an interesting approach to a looming global food concern; methods are clearly outlined and the results are effectively displayed.  As pointed out in their limitations section, their results “may prove difficult to reach in practice” as a large shift away from meat consumption is required concomitant with seemingly impractical increases in plant-based alternative foods.  An unintended consequence of this approach, however, is not adequately presented in the discussion:  the fact that protein intake falls 15-25% (to approximately 75-80 g/d).  Given that plant proteins are less available (lower digestibility and bioavailability) than animal protein, this reduction in protein intake is an important health consideration that deserves mention. 

Author Response

We thank the reviewer for taking the time to review our manuscript and for his/her valuable suggestions.

We agree that this fact has been overlooked in the results and discussion, and therefore have addressed this point in the Results and Discussion (line 352-360).